# How Multilingual Are Large Language Models Fine-Tuned for Translation?

**Aquia Richburg**
AMSC
University of Maryland
College Park, MD 20742, USA
`arichbu1@umd.edu`

**Marine Carpuat**
Computer Science
University of Maryland
College Park, MD 20742, USA
`marine@umd.edu`

## Abstract

A new paradigm for machine translation has recently emerged: fine-tuning large language models (LLMs) on parallel text has been shown to outperform dedicated translation systems trained in a supervised fashion on much larger amounts of parallel data (Xu et al., 2024a; Alves et al., 2024). However, it remains unclear whether this paradigm can enable massively multilingual machine translation or whether it requires fine-tuning dedicated models for a small number of language pairs. How does translation fine-tuning impact the MT capabilities of LLMs for zero-shot languages, zero-shot language pairs, and translation tasks that do not involve English? To address these questions, we conduct an extensive empirical evaluation of the translation quality of the TOWER family of language models (Alves et al., 2024) on 132 translation tasks from the multi-parallel FLORES-200 dataset. We find that translation fine-tuning improves translation quality even for zero-shot languages on average, but that the impact is uneven depending on the language pairs involved. These results call for further research to effectively enable massively multilingual translation with LLMs.

## 1 Introduction & Background

Machine translation (MT) systems have long been built for a specific pair of input and output languages, and trained on a parallel corpus representing this language pair and translation direction (Brown et al., 1990). End-to-end neural models made it easier to train a single model on multiple language pairs, enabling multilingual machine translation models (Johnson et al., 2017; Arivazhagan et al., 2019b; Dabre et al., 2020; Team et al., 2022), where low-resource languages benefit from transfer learning from high-resource languages, including in zero-shot translation directions. While transformative, this paradigm still assumes access to large amounts of parallel data which is unrealistic for many of the world's languages – less than 5% of languages have labeled data useful enough for applications (Joshi et al., 2020).

Over the past year, large language models (LLMs) have started to exhibit competitive translation quality compared to dedicated supervised MT systems (Kocmi et al., 2023). While LLMs have been used for translation in zero-shot and few-shot settings (Vilar et al., 2023; Hendy et al., 2023; Zhang et al., 2023a; Bawden & Yvon, 2023; Lin et al., 2022; Zhu et al., 2023, among others), the performance of LLMs often lagged behind dedicated MT systems trained in a supervised fashion on parallel text, particularly for low-resource languages. Instruction fine-tuning of a multilingual LLM for MT has emerged as a strong contender to build state-of-the-art systems by providing a straightforward mechanism to combine the strengths of multilingual pre-training on primarily monolingual data with the controlled injection of parallel data – rather than only relying on incidental bilingualism (Briakou et al., 2023). Xu et al. (2024a) showed that fine-tuning LLAMA2 models in two stages – first on monolingual data followed by small amounts of high-quality parallel data – could outperform dedicated supervised MT systems (Team et al., 2022) and large closed models, like GPT3.5, with 7B or 13B parameter open LLAMA2 models. Translation quality has been

further improved with contrastive preference optimization (Xu et al., 2024b). Meanwhile, Alves et al. (2024) recently introduced a family of LLM for translation-related tasks which is also based on LLAMA2. It involves continued pre-training on monolingual and parallel text, as well as fine-tuning on parallel text (and related tasks such as post-editing). The resulting models exhibit impressive machine translation performance, outperforming dedicated multilingual models (Team et al., 2022) and Xu et al. (2024b)'s best models, when translating between English and the non-English languages covered in the training set. In the same vein, recent models such as BigTranslate (Yang et al., 2023) or BayLing (Zhang et al., 2023b) employ continued pre-training on parallel data or translation instruction fine-tuning to improve the translation capabilities of LLMs.

However, it remains unclear how well LLMs fine-tuned for MT translate languages that they have not been trained on. Multilingual LLMs are imbalanced in their language coverage and are affected by the curse of multilinguality during pre-training (Chang et al., 2023; Liang et al., 2023); this is compounded by the fact that exposure to fine-tuning parallel data is necessarily limited to a small number of language pairs raising the same questions about maximizing the benefits of transfer learning for low-resource languages without degrading performance for high-resource languages that have been extensively studied in multilingual MT (Arivazhagan et al., 2019a; Liu & Niehues, 2022; Chang et al., 2023; Yuan et al., 2023, among many others).

To fill this gap, this work seeks to characterize how LLMs fine-tuned for translation perform across a diverse set of translation tasks beyond the ones they were fine-tuned on. Addressing this question is crucial to determine whether using LLMs for MT will require fine-tuning models for individual (or small sets of) language pairs, or whether they can truly fulfill the promise of multilingual LLMs and enable massively multilingual MT. We present an extensive empirical evaluation of the TOWER models on 132 machine translation tasks with varying degrees of supervision. We contribute an experimental design and result analysis inspired by Choudhury & Deshpande (2021)'s critique of averaging performance over a set of languages as the primary selection rationale for massively multilingual LLMs: we aim to balance the need to cover an extensive set of languages to understand the generalization ability of the models, with the desire to understand performance beyond a single score which averages potentially disparate behaviors across languages. Our selection of languages is driven by both linguistic typology and data size criteria. Furthermore, we include translation between all language pairs, and do not limit ourselves to translation into and out of English. While MT research naturally does not focus on English as much as other Natural Language Processing research (Bender, 2019; Joshi et al., 2020), the vast majority of MT evaluations involve English as either the source or target language. Here we also seek to assess to what degree systems trained on English-centric parallel data generalize to translation tasks that do not involve English.[1]

To be explicit, this work does not introduce any new modeling techniques. Instead, we contribute:

- An extensive empirical evaluation of the generalization ability of LLMs fine-tuned for translation, using TOWER models on MT tasks involving 132 translation directions between 12 languages, including German (De), English (En), Korean (Ko), Dutch (Nl), Russian (Ru) and Chinese (Zh) – the supervised languages seen during fine-tuning – and Czech (Cs), Icelandic (Is), Japanese (Ja), Polish (Pl), Swedish (Sv) and Ukrainian (Uk) – the zero-shot languages not seen during fine-tuning (Section 2).

- A fine-grained analysis of results that goes beyond averaging scores across languages and considers best and worst behavior per model and per source and target language to provide a more complete picture of model performance across languages (Choudhury & Deshpande, 2021) (Section 3).

---

[1]This goal is shared with Gao et al. (2024), however, their evaluation is limited to language pairs that involve languages seen at fine-tuning time, while we also consider generalization to zero-shot *languages* in addition to zero-shot *language pairs*.

| Language | Script | LLaMA-2 support | Similarity Groups |
|---|---|---|---|
| Czech | Latin | 0.03% | West Slavic |
| Polish | Latin | 0.09% | West Slavic |
| **Russian** | Cyrillic | 0.13% | East Slavic |
| Ukrainian | Cyrillic | 0.07% | East Slavic |
| **German** | Latin | 0.17% | West Germanic |
| **English** | Latin | 89.70% | West Germanic |
| Icelandic | Latin | possibly unseen | North Germanic |
| **Dutch** | Latin | 0.12% | West Germanic |
| Swedish | Latin | 0.15% | North Germanic |
| Japanese | Kana | 0.10% | Kanji from Hanzi, SOV order |
| **Korean** | Hangul | 0.06% | SOV order |
| **Chinese** | Hanzi | 0.13% | Hanzi to Kanji, loanwords to Ja and Ko |

Table 1: Evaluated languages with rationales for similarity grouping. The languages marked in **bold** belong to the supervised set for both TOWER models.

- Findings that indicate that fine-tuning on translation-related tasks improve LLM's ability to handle the task of translation itself beyond the specific language pairs seen during fine-tuning (Section 3).

- However, the fine-grained analysis reveals an uneven picture with higher variance in translation quality for language pairs that are not fully supervised, and low quality in outlier languages whether they are seen during fine-tuning (Korean) or not (Icelandic) (Section 3).

## 2 Evaluation Design

**Evaluation Languages**    Since we are interested in measuring the supervised and zero-shot translation capabilities of LLMs before and after fine-tuning, we use a subset of the fine-tuning languages from TOWER as our supervised set and a collection of related languages seen as some proportion of LLAMA2's pre-training data as our zero-shot set, as summarized in Table 1. The **supervised language set** consists of German (De), English (En), Korean (Ko), Dutch (Nl), Russian (Ru) and Chinese (Zh). The **zero-shot language set** consists of Czech (Cs), Icelandic (Is), Japanese (Ja), Polish (Pl), Swedish (Sv) and Ukrainian (Uk). The languages in the zero-shot set are chosen to represent a range in resource support and to be related to languages in the supervised set by language family, typological properties or orthography. This yields $\binom{12}{2}$ = 66 pairs from the set of languages, and thus **132 translation tasks** when considering both translation directions. We divide these 132 tasks into 3 subcategories based on the amount of supervision available in TOWER:

- **Fully supervised translation directions**: These consist of De, Ko, Nl, Ru and Zh to and from En. This totals 10 pairs.

- **Partially supervised translation directions**: These are the pairs where at least one language is from the supervised set, but the exact language pair was not seen during fine-tuning. For example, De-Zh belongs to this set because no direct De-Zh parallel data was used during fine-tuning. This totals 92 pairs.

- **Unsupervised translation directions**: These are the pairs where both languages are in the zero-shot set. This totals 30 pairs.

**Test Data & Metrics**    For all experiments, we evaluate on the multilingual, multi-parallel FLORES-200 (Team et al., 2022) devtest set, which consists of English language articles manually translated into multiple languages. We report reference-based COMET-22 (Rei et al., 2022a) as our main metric for translation quality. We also computed the commonly

used BLEU (Papineni et al., 2002) metric, however in addition to its well-documented weaknesses, it exhibits a higher variance over diverse languages due to the differences in tokenization.

**LLMs Translation Models** Our zero-shot baseline model is MetaAI's LLAMA2 (Touvron et al., 2023), an LLM which has been shown to be a strong baseline on a number of English language tasks. For fine-tuning, we use the TOWER (Alves et al., 2024) family of models, which fine-tunes LLAMA2 for a range of translation-related tasks. TOWERBASE uses a two-third/one-third mixture of monolingual and parallel data to continue pre-training LLAMA2. The monolingual data is sourced from mC4 (Xue et al., 2021), a multilingual webcrawl corpus, and the parallel data comes from several public sources ranging in domain from news to medical to Wikimedia. TOWERINSTRUCT further optimizes TOWERBASE for translation through instruction fine-tuning, formatting the translation task as natural language instructions. A diverse collection of data resources are formulated into question-answer templates including some zero-shot directions. TOWERINSTRUCT uses data with more task diversity including paraphrasing, dialog data and coding instructions. We consider both the 7B and 13B parameter model variants for each of the LLMs. At inference time, for the TOWER models, we use the same prompt format as presented during fine-tuning. Since LLAMA2 was not trained for translation, we default to a basic natural language prompt to translate the source sent from Language A to Language B. We use beam search with a beam-width of 5 and a maximum of 128 new tokens throughout.

**Dedicated MT system** We also include the state-of-the-art No Language Left Behind (NLLB) model Team et al. (2022).[2] NLLB is a supervised multilingual NMT system trained to translate between a diverse variety of 200 languages, including using parallel data that does not include English. Among the languages we evaluated, all language pairs that involve English and Korean ↔ Japanese are seen at training time. At inference time, we use language specific tokens marking the source and target consistent with NLLB training and use the same beam search setting as for the LLMs.

**Limitations** While our empirical study is extensive, it comes with inherent limitations. Our use of the FLORES-200 test data means that we use text that was originally written in English and translated into other languages, no matter what MT direction is evaluated, which might introduce translationese effects (Team et al., 2022). We accept this limitation as the multi-way parallel nature of FLORES-200 has the advantage of letting us make more controlled comparisons across languages. Our evaluation is based on automatic metrics that are known to correlate highly with human judgments Rei et al. (2022b), it remains to be seen how the translation quality differences we observe are perceived by people.

## 3 Results

**Impact of Supervision Type** In Figure 1, we plot the distribution of COMET scores for each model according to translation supervision type. The dedicated NLLB MT model represent a strong baseline across supervision types, with mean COMET scores of 0.8 or higher. As can be expected, the LLM systems reach higher mean COMET scores for the fully supervised language pairs than others. When comparing the LLMs with the dedicated NLLB system, we find that the zero-shot LLAMA2 models lag behind by a wide margin. Interestingly, the smaller LLAMA2-7B model yields higher mean COMET scores than the larger LLAMA2-13B model. For supervised language pairs, all TOWER models are on par or better than NLLB. However, their performance is more uneven in the other supervision conditions: continued pre-training in the TOWERBASE models improves COMET scores compared to the zero-shot LLAMA2 results but the mean COMET scores lag behind those of the dedicated NLLB systems. Translation-related fine-tuning in TOWERINSTRUCT models improves translation quality further, but only the larger TOWERINSTRUCT-13B yields higher

---

[2]We use the distilled 600M to maximize inference speed. On a more limited set of language pairs, Alves et al. (2024) report consistent results with the larger 54B variant.

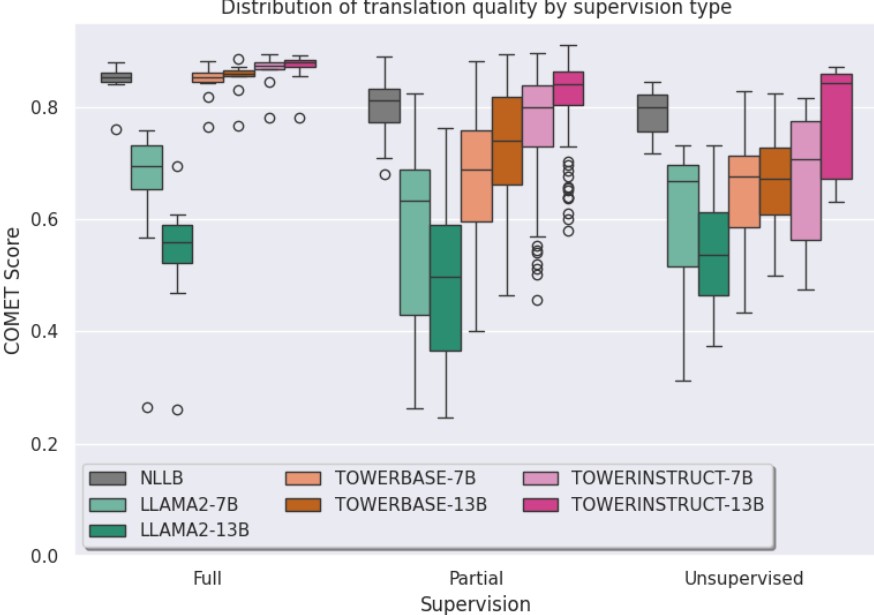

Figure 1: Distribution of COMET scores for each translation approach over language pair supervision type: TOWERINSTRUCT-13B models are competitive with NLLB on average across settings, suggesting that transfer learning benefits many unsupervised languages, but not all based on the increased variance in the least supervised conditions.

mean COMET scores than NLLB. These results suggest the translation supervision at pre-training and fine-tuning time does improve LLM's translation abilities beyond the languages seen in parallel text, and thus that transfer learning benefits other languages.

However, the mean scores only provide a partial picture of the translation quality across the 132 language pairs considered. Figure 1 clearly shows, for all models, that the variance of COMET scores across languages also increases in the partial supervision and unsupervised settings compared to the fully supervised settings, which motivates us to look at results in a finer-grained fashion. The variance increases more for LLM than NLLB models indicating that the dedicated NLLB models have a more reliable behavior across language pairs. TOWERINSTRUCT has a large tail of low quality outliers, some of which are lower than the mean zero-shot LLAMA2 scores. This shows that translation-related fine-tuning does not benefit all language pairs uniformly.

**Best/Worst Translation Paths per Model** To dig deeper, we examine differences in translation quality across languages for fixed models. Figure 2 represents the spread of COMET scores for each target language for the dedicated NLLB model (top) and the overall top-performing TOWERINSTRUCT-13B (bottom).[3] Results show that while on average TOWERINSTRUCT-13B is the top performing model, its worst performance is consistently worse than that of NLLB, for both supervised and unsupervised target languages. Translation from Icelandic almost always yields the worst COMET score no matter the target language (red points). The best performing dissimilar translation paths (D+ points) interestingly tend to overlap with other top performing languages, indicating that dissimilarity between source and target language is not necessarily a challenge for TOWERINSTRUCT-13B.

---

[3]The complete set of graphs for all models and source/target language groupings can be found in Appendix C.

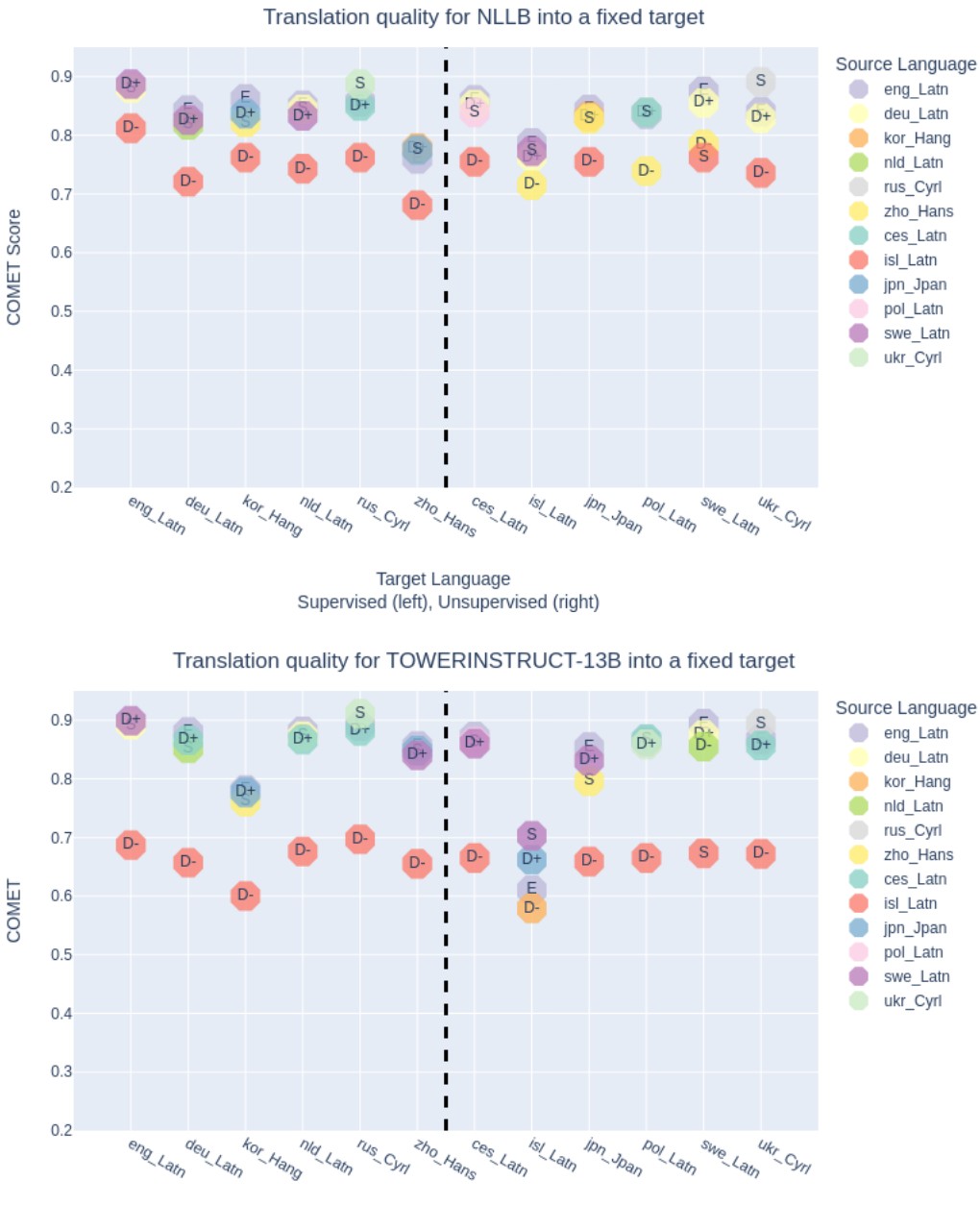

Figure 2: Spread of translation quality as measured by COMET score for each target language for the dedicated NLLB model (top) and the overall top-performing TOWERINSTRUCT-13B (bottom). For each data point, the color identifies the source language, and we mark the best/worst dissimilar paths (D+/D-), the similar path (S), and translation from English (E).

**Best/Worst Models per Language Pair** Next, we show the models with the absolute best and worst translation quality per language pair in Figure 3. TOWERINSTRUCT-13B emerges as the top performing model across language pairs (top graph), while the zero-shot LLAMA2 models often yield the worst performance (bottom graph); however there are interesting exceptions in both cases.

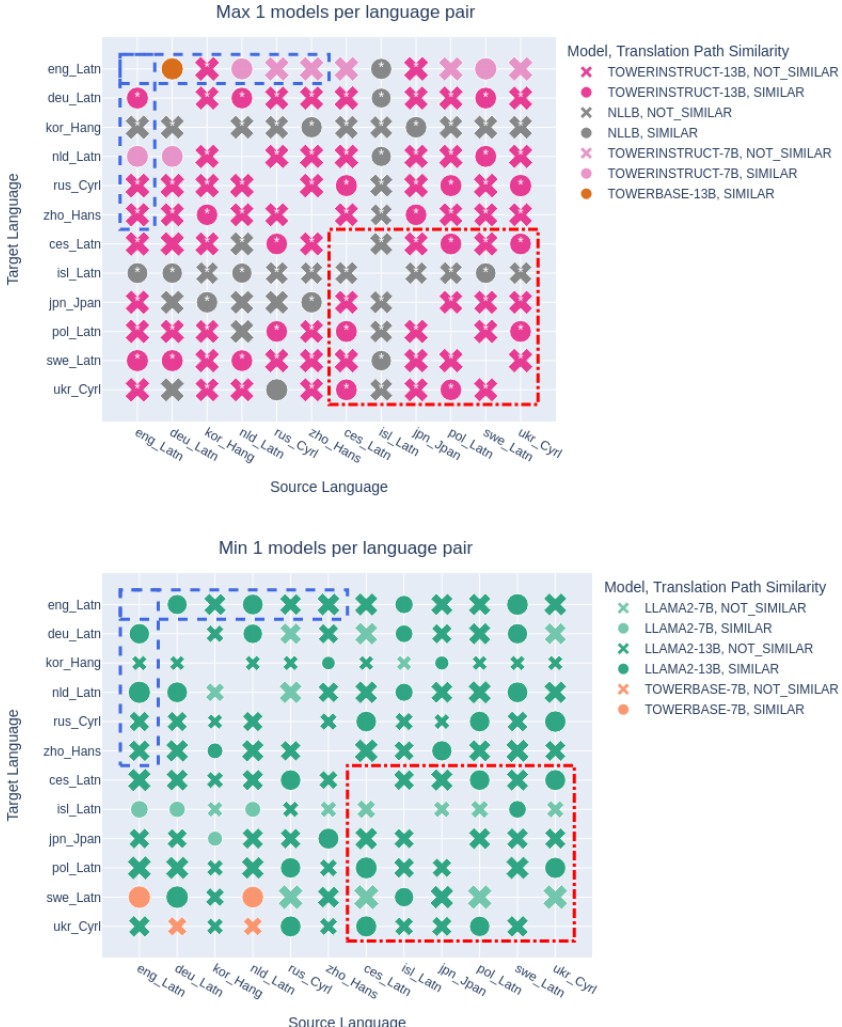

Figure 3: Models that achieve the absolute best and worst scores per language pair. The group outlined in dotted-blue corresponds to the fully supervised set, the group outlined in dot-dash-red corresponds to the unsupervised set and the remaining pairs are partially supervised. The circle and cross symbols denote whether the language pair consists of similar or dissimilar language, respectively (as defined in Table 1). Symbols marked with asterisks denote statistical significance according to a paired *t*-test at a significance level of 0.05.

Among the best performing systems, we see that for translation into English and for supervised languages, the largest TOWERINSTRUCT-13B is not necessarily best: the smaller TOWERINSTRUCT-7B yields higher COMET scores for translation from 5 language pairs into English (top graph, top row), and surprisingly the TOWERBASE-13B model outperforms the fine-tuned TOWERINSTRUCT models for translation from German into English. So bigger models with more supervision are not uniformly better. Furthermore, LLMs all struggle compared to NLLB with translation involving Icelandic and Korean. When Korean is the source TOWERINSTRUCT wins 9 out of 11 times but loses completely when Korean is switched to the target. In the case of Icelandic, TOWERINSTRUCT loses in both directions. Icelandic is not seen at fine-tuning time, accounts for 1% at most of LLAMA2 pre-training data, and is a fusional language, so it is perhaps unsurprising that is challenging to translate for the LLMs. The Korean results are more surprising since it is included in fine-tuning. However,

its typography (unique Hangul) and grammar (Subject-Object-Verb) differ vastly from the other included supervised languages (mainly Subject-Verb-Object) which might make it harder to model on the target side and explain why NLLB remains stronger. This suggests that when selecting a set of fine-tuning languages, more typological diversity might support better generalization to new languages.

Interestingly, the worst performing systems are often the larger LLAMA2-13B zero-shot models, except for translation into Korean and Icelandic where the smaller LLAMA2-7B zero-shot models are worse. This suggests that the increased capacity in the underlying model benefits the representation of these outlier languages, but not for the more similar higher-resource languages. Additionally, for a small number of unsupervised language pairs (English-Swedish, German-Ukrainian, Dutch-Ukrainian and Dutch-Swedish), continued pre-training hurts translation quality and TOWERBASE-7B models underperform the zero-shot LLAMA2 models. Similar versus dissimilar translation directions do not show obviously different patterns.

To understand these results better, we turn to examining the properties of the inputs and outputs of NLLB versus LLM translations.

**Off-Target Outputs**    A known failure mode of LLM is to generate translations in the wrong target language: for instance, Xu et al. (2024a) note that this behavior for LLAMA2 when translating out of English. One possible explanation is that when faced with translation on languages with lower support the model defaults to a higher-resource language, usually English. We report the average percentage of translations that are found not to match the specified target language for our LLMs using the lingua-py[4] language identification tool in Figure 4. The analysis of off-target translation across the multiple language paths considered here are consistent with prior work, zero-shot LLAMA2 models are often off target. Interestingly the larger LLAMA2-13B is more often off-target than the smaller LLAMA2-7B model, which contributes to its worse overall performance (Figure 1). Continued pre-training and fine-tuning substantially improve the on-target behavior of supervised languages (Figure 4 left), however off-target outputs remain a problem even after fine-tuning when translating into unsupervised languages (Figure 4 right). These results suggest that LLM translations into Korean and Icelandic are bad for different reasons: translation into Icelandic is only on target about 40% of the time, while translation into Korean is generally on target (more than 90% of the time).

**Input Tokenization**    Based on the typological property of each language and its pre-training representation, parallel texts can be encoded into subword sequences of vastly different lengths across languages, leading to discrepancies in the quality and cost of inference across languages (Limisiewicz et al., 2023; Ahia et al., 2023; Petrov et al., 2023). The tokenizers for NLLB and our LLM models are all built using SentencePiece (Kudo & Richardson, 2018). NLLB trains a 256k vocabulary model by temperature sampling 100M bitext pairs to downsample higher resource languages and upsample lower resource languages (Team et al., 2022). The LLAMA2 tokenizer (also used by the TOWER models) has a vocabulary size of 32k trained on a mostly English data mixture (Touvron et al., 2023). Figure 5 plots the average number of source subwords per segment (top) in parallel with the average COMET scores for all models for translation out of each source language (bottom).[5] The NLLB and LLAMA2 tokenizers both favor English, with inputs in all languages being longer than English indicating over-segmentation. The LLM tokenizer over-tokenizes at a higher rate than the NLLB tokenizer, and has wider discrepancy across languages: Korean text is more than 3 times longer than when translating into English, which might be a factor in the lower translation quality of TOWERINSTRUCT-13B into Korean compared to other language pairs, despite its supervised status. Icelandic inputs are more than twice longer than English, but are somewhat shorter than Japanese inputs which are better translated by the LLM models. Input length thus does not entirely explain translation quality.

---

[4]https://github.com/pemistahl/lingua-py
[5]We provide the symmetric figure for output length in Appendix Figure 7.

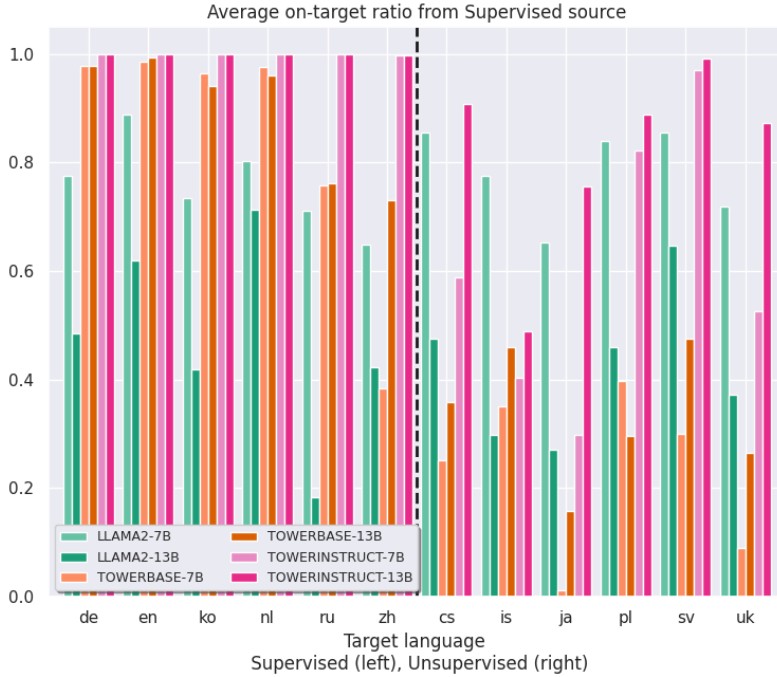

Figure 4: Ratio of on-target translation averaged over the source sentences from the supervised set. The TOWERINSTRUCT models are generally on target for supervised languages, but off-target translations are frequent for unsupervised languages. By contrast, NLLB translations are on target for 97-100% of outputs.

## 4 Conclusion

This work sought to characterize how LLMs fine-tuned for translation perform across a diverse set of translation tasks beyond the ones they were fine-tuned on. By conducting an extensive empirical evaluation of the TOWER family of LLMs on 132 translation tasks representing diverse degrees of supervision, we find that fine-tuning on translation-related tasks improves LLM's ability to handle the task of translation itself beyond the specific language pairs seen during fine-tuning. This encouragingly suggests that the fine-tuning paradigm has the potential to enable massively multilingual MT.

However, the fine-grained results show that transfer learning has an uneven impact: outlier languages remain hard to handle whether they are seen during fine-tuning (Korean) or not (Icelandic). Analysis suggest that Korean might be harmed by oversegmentation compared to other fine-tuning languages, while translation involving Icelandic often results in off-target outputs. Furthermore, the worst-case behavior of the best LLM (TOWERINSTRUCT-13B) is consistently worse than that of the dedicated NLLB model. This highlights the need for future work on improving instruction-tuning techniques to benefit the harder language pairs. Initial efforts in that direction are underway. For instance, Gao et al. (2024) have ported a cross-lingual consistency regularization method developed for dedicated multilingual MT (Gao et al., 2023) to instruction fine-tuning – while promising when tested on languages seen during fine-tuning, its impact on translation involving languages unseen during fine-tuning remains to be seen. Our results further suggest that ensembling methods that back-off to dedicated MT models might be useful to mitigate off-target translations, and also call for addressing tokenization fairness as a potential cause for discrepancies in translation quality across languages.

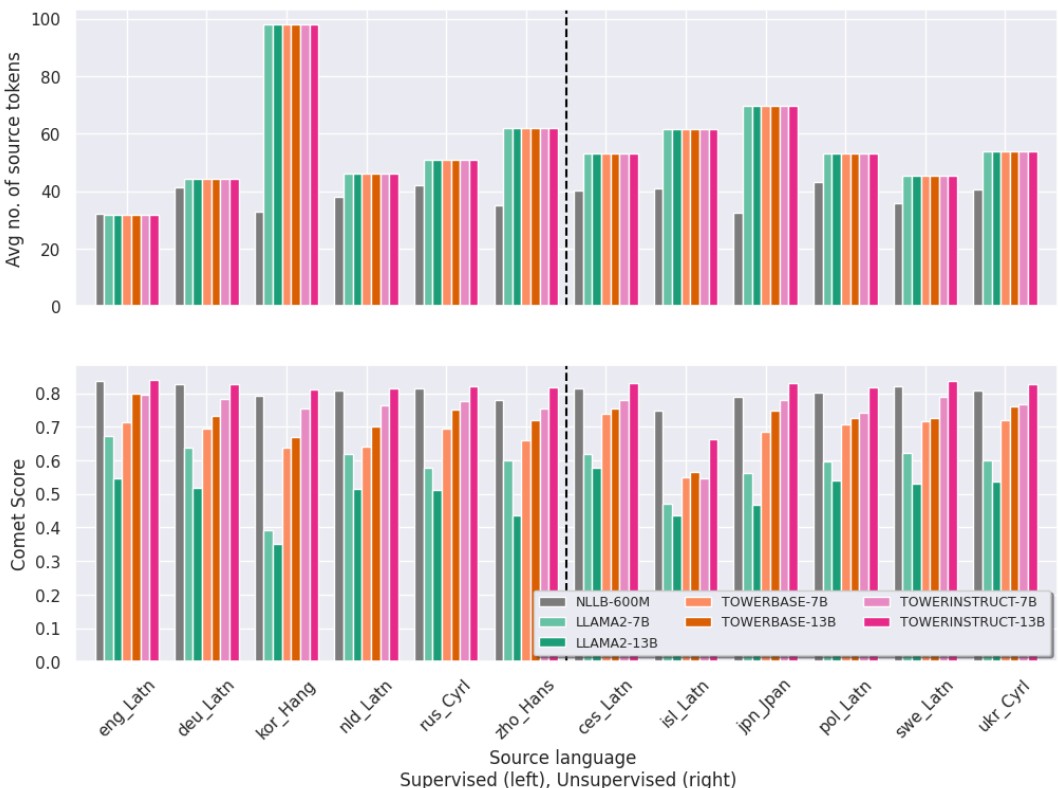

Figure 5: Average input length per language after subword segmentation for each model (top) and average translation quality out of each source language per model (bottom).

While we have studied overall translation quality as measured by COMET, other aspects of the outputs would be worth studying in future work including how translationese effects (Dutta Chowdhury et al., 2020; Vanmassenhove et al., 2021) and hallucination patterns (Guerreiro et al., 2023) are impacted by different degrees of instruction-tuning supervision.

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

## A   Appendix: Other Metrics for Translation Quality

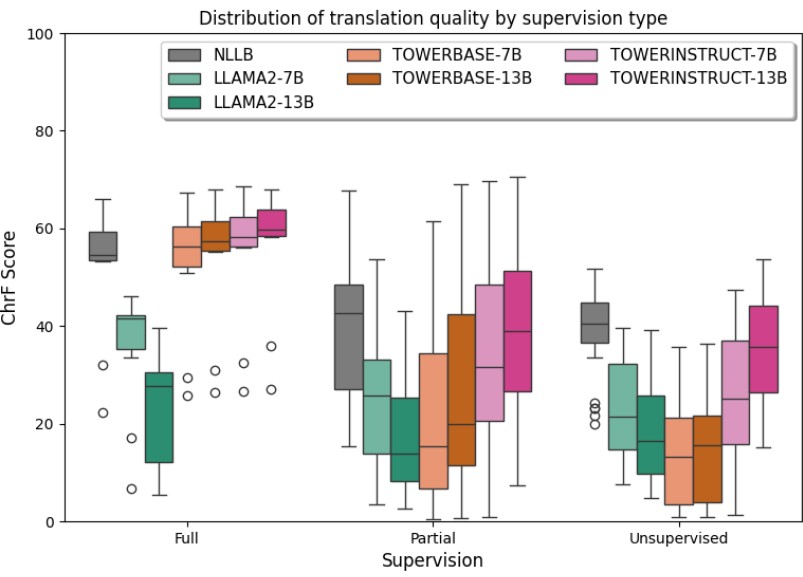

Figure 6: Distribution of ChrF scores for each translation approach over language pair supervision type.

We plot the distribution of ChrF scores in Figure 6. We generally see a similar trend where fine-tuning shows improvement across all supervision types. The range in scores is wider compared to COMET. This indicates that translation outputs vary to a degree from the reference.

## B   Appendix: Impact of Target Tokenization

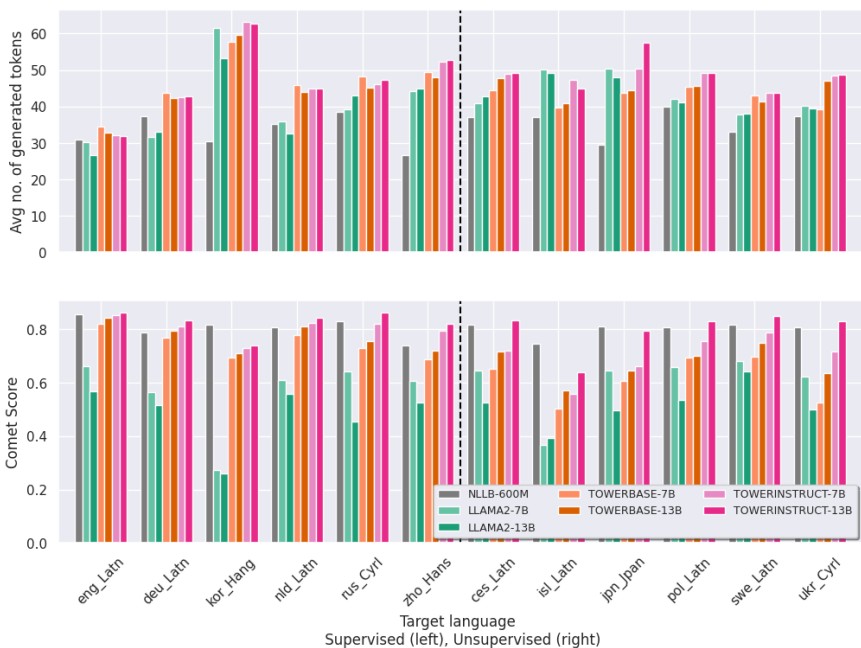

Figure 7: Average translation quality into target and average number of generated tokens in target.

## C   Appendix: Translation Quality per Model and Source/Target Language

We illustrate the spread of translation quality scores for all models and source or target languages in Figures 8 to 12 below, to complement Figure 2 in the main paper which show this spread per target languages, for the NLLB and TOWERINSTRUCT-13B models only.

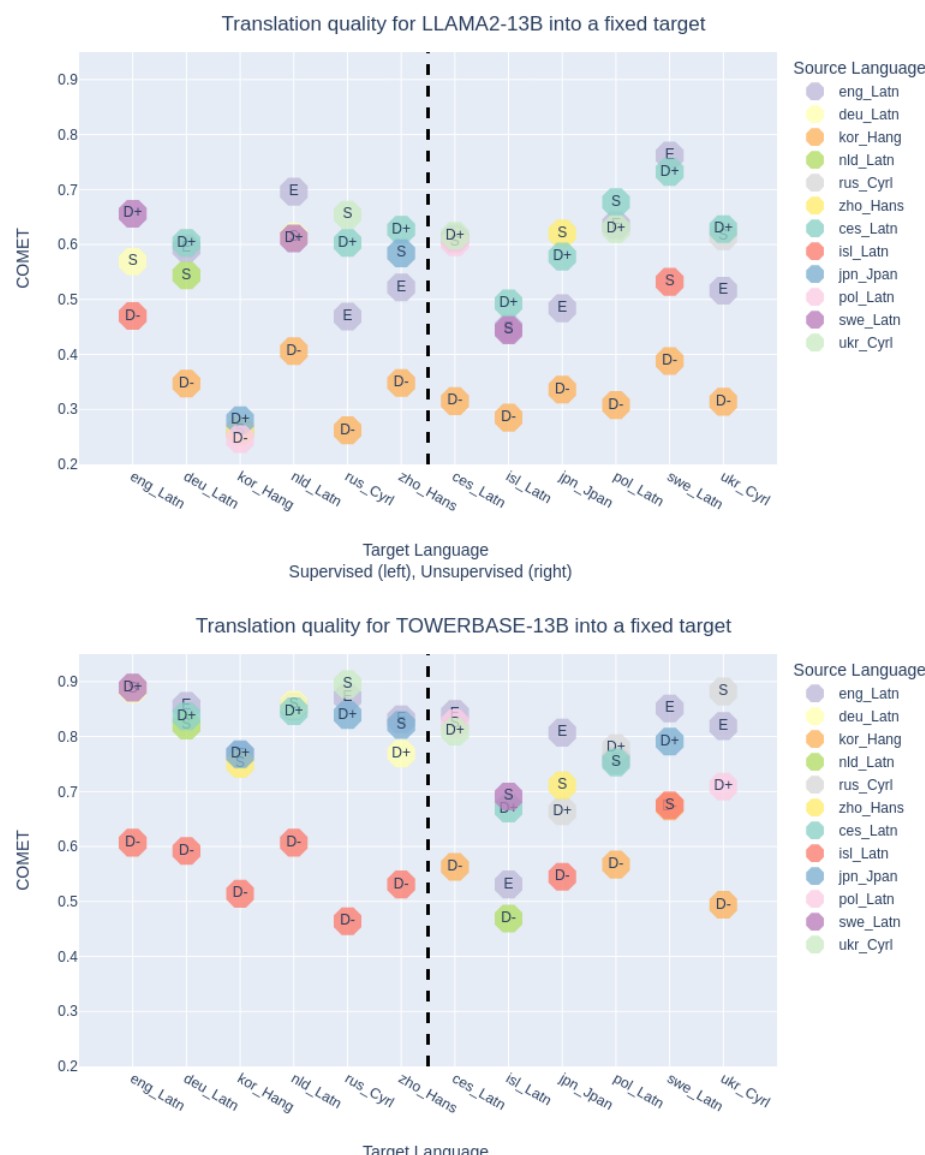

Figure 8: Spread of translation quality as measured by COMET score for each target language for the zero-shot LLAMA2-13B model (top) and the monolingual/bitext fine-tuned TOWERBASE-13B (bottom). For each data point, the color identifies the source language, and we mark the best/worst dissimilar paths (D+/D-), the similar path (S), and translation from English (E).

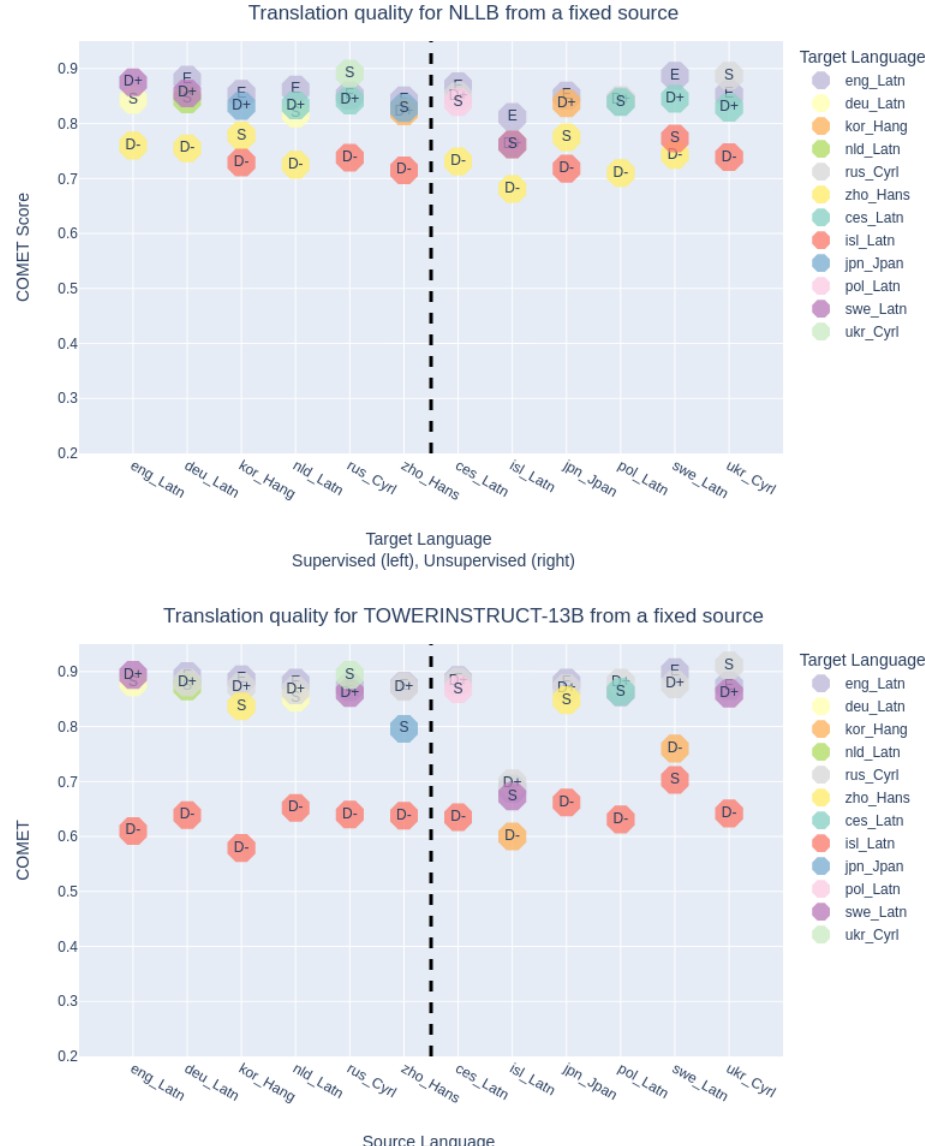

Figure 9: Spread of translation quality as measured by COMET score from each source language for the dedicated NLLB model (top) and the overall best performing TOWERINSTRUCT-13B (bottom). For each data point, the color identifies the target language, and we mark the best/worst dissimilar paths (D+/D-), the similar path (S), and translation from English (E).

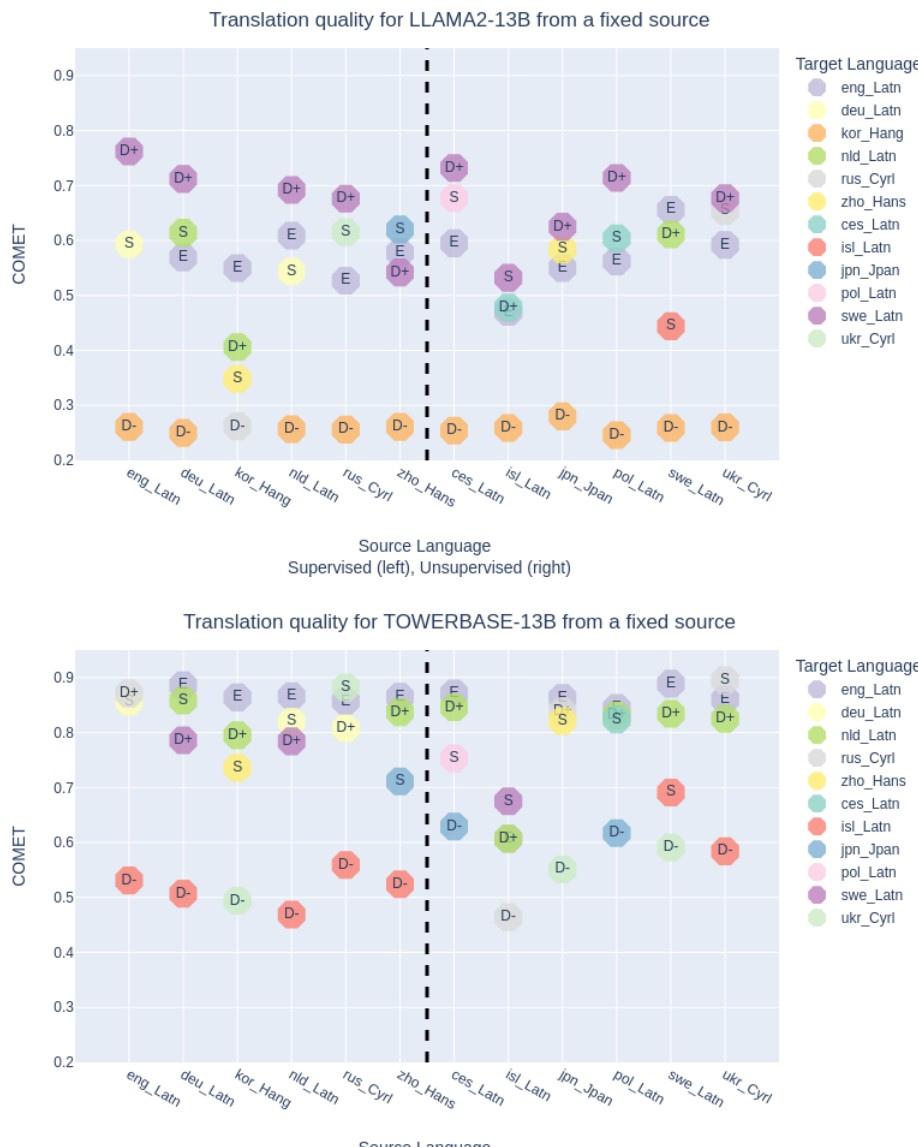

Figure 10: Spread of translation quality as measured by COMET score from each source language for the zero-shot LLAMA2-13B model (top) and the monolingual/bitext fine-tuned TOWERBASE-13B (bottom). For each data point, the color identifies the target language, and we mark the best/worst dissimilar paths (D+/D-), the similar path (S), and translation from English (E).

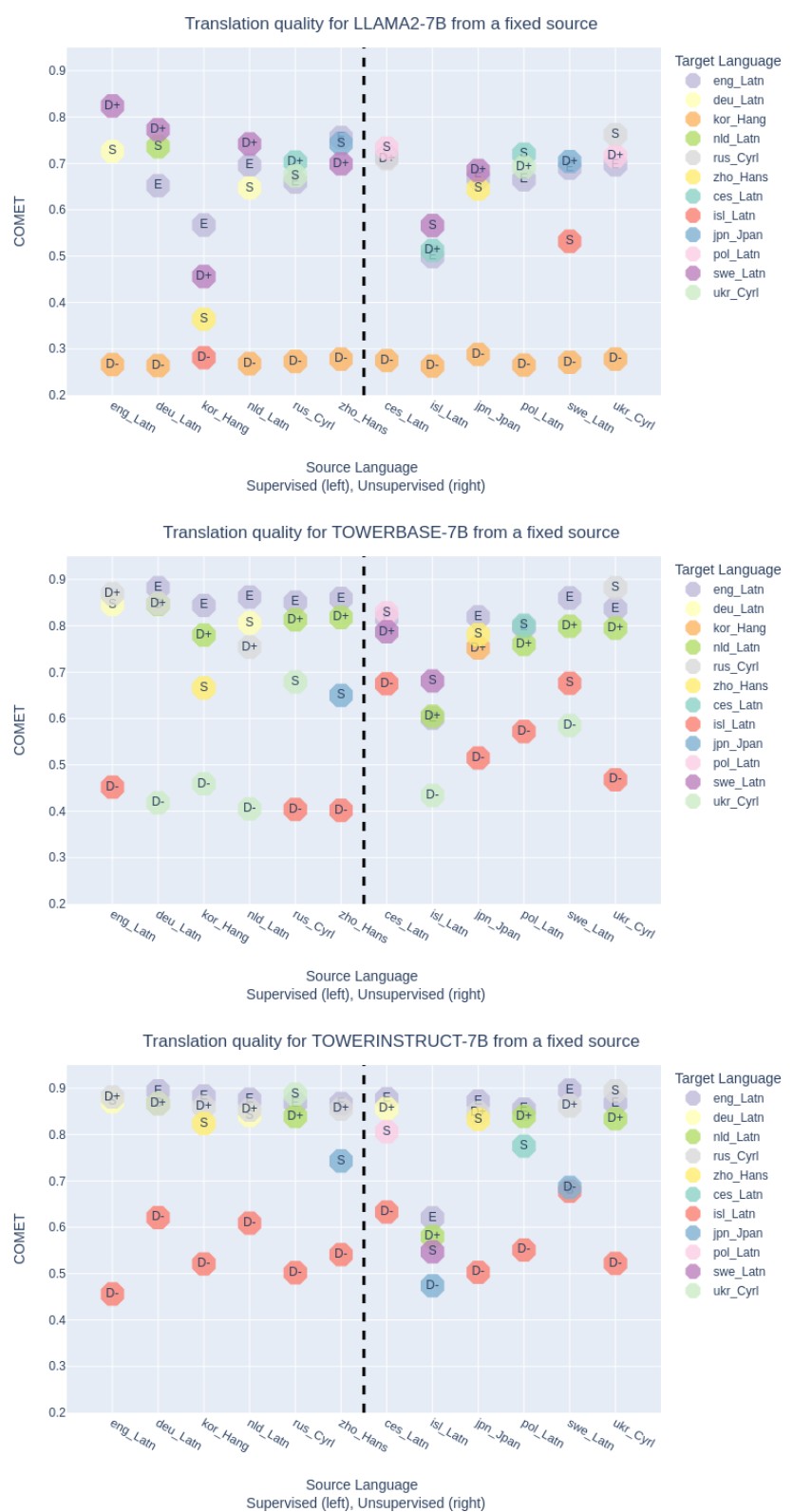

Figure 11: FROM SOURCE; S: similar, E: English, D+: best dissimilar, D-: worst dissimilar

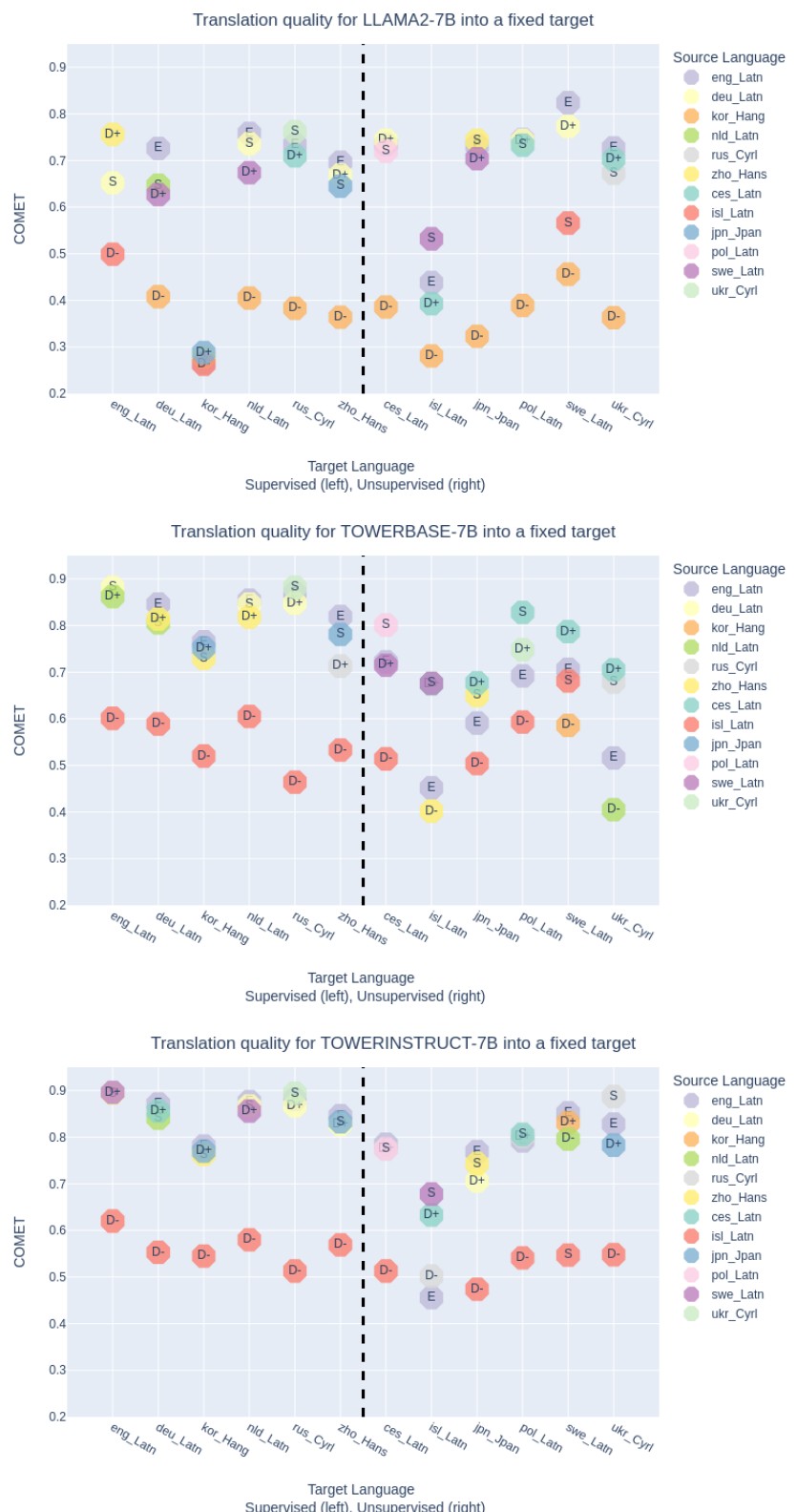

Figure 12: TO TARGET; S: similar, E: English, D+: best dissimilar, D-: worst dissimilar

