# OpenReview forum: "How Multilingual are Large Language Models Fine-tuned for Translation?"
_colmweb.org/COLM/2024/Conference — COLM_

### Official Review · Reviewer_sAju · 2024-05-02

**Rating:** 7
**Confidence:** 4
**Ethics Flag:** 1

**Summary:**

This paper investigates the capabilities of a large language model (LLM) fine-tuned for machine translation (MT) on certain languages to perform across various language directions for which it has not been tuned. This paper does not present any technical novelty but tries to understand better the capabilities of an LLM challenging it in unseen language directions. The experiments include 132 language directions split into fully supervised, partially supervised, and unsupervised directions, depending on the data used during training. When dealing with a large number of directions, how to report the results is an important problem, because the widely-used average does not allow a reader to highlight the positive and negative performance of a model. This paper addresses this problem by considering the best and worst behaviors of a model and linking them with the source and target languages. A large set of experiments with in-depth analysis are reported to verify the initial research questions.

The main findings of the paper are:

-) The fine-tuning of an LLM for translation-related tasks helps to improve the performance also on unseen languages.

-) The paper identifies high variance in translation quality for language pairs that are not supervised and outliers languages such as Icelandic and Korean.

**Questions To Authors:**

Do the authors have any insights on the impact of the alphabet used by each language on the translation performance? And what is the role of the vocabulary size (smaller vocabulary can penalize those languages with a high-variable alphabet or an alphabet with a large set of specific symbols)?

**Reasons To Accept:**

Although it does not propose any technical enhancement, the paper addresses an interesting problem that should help to understand better the behaviors of LLMs for translation. In this historical period, where there is an increasing emphasis on improving LLM performance on multiple tasks, these papers are important to have a critical view of the existing approaches and provide some directions to work on.

The paper is supported by a large set of experiments that include results for high- and low-resourced languages. The proposed analysis has clear questions that are addressed with proper experiments.

The paper is clear and easy to follow.

**Reasons To Reject:**

The findings are clear but not conclusive leaving more activities to better understand the capabilities of the LLMs for translation and to motivate the outlier behaviors of some languages.

The proposed method for overcoming the average limitation when reporting results for a large number of pairs is interesting, but it may not be immediately comprehensible and may require some effort to fully appreciate. However, this is the trade-off for introducing a new metric or visualization strategy.

The experimental parts of the paper are not always easy to follow and appreciate. The fact that the figures and tables are not on the page where they are discussed creates more difficulties.

Overall the paper is well-written, but it lacks the final proofreading. Some sentences to adjust:

-) Page 1: “Briakou et al. (2023). vast amounts of noisy multilingual pre-training data. Xu” not clear how the sentence starting with “vast” is connected to the previous and next sentences.

-) Page 3: the table with the language information is table 1 and not table 2.

-) Page 5: “for each target language for the for the dedicated” “For the” is repeated.

-) Page 8: “upsample lower resource languages .“ extra space to be removed.

---

> ### Author Rebuttal · Authors · 2024-05-30
>
> Thank you for your comments on the paper.
> We have made the editing fixes suggested.  Thank you!
>
> While we do not directly study the impact of writing script, we report how the different amounts of tokenization across languages impact translation quality (in Figure 5), which is impacted by the nature of the script.  The LLAMA tokenizer used by the TOWER models is mainly trained on English text and other text using variants of the Latin alphabet.  And we see translation with the European languages are generally higher than with the Asian languages (in Figure 2, for best performing TOWER variant).  In contrast, NLLB has a more even representation of writing scripts among the language collection and results have less disparity.  It remains to be seen whether a more equitable tokenizer could improve translation quality for the LLM.

---

> > ### Comment · Reviewer_sAju · 2024-06-03
> > **Acknowledge**
> >
> > Thanks a lot for the clarification

---

### Official Review · Reviewer_zRXH · 2024-05-11

**Rating:** 6
**Confidence:** 3
**Ethics Flag:** 1

**Summary:**

The paper conducts an extensive empirical evaluation of the translation quality of the TOWER family of language models on 132 translation tasks from the multi-parallel FLORES data, aiming to find out how does translation fine-tuning impact the MT capabilities of LLMs for zero-shot languages, zero-shot language pairs, and translation tasks that do not involve English. The conclusion is interesting that the translation fine-tuning improves translation quality even for zero-shot languages on average.

**Questions To Authors:**

It will be better if the finding can be further confirmed on other LLMs.

**Reasons To Accept:**

1. The paper addresses an interesting problem about the multilingual translation ability of LLM models.
2. Empirical experiments show that the translation fine-tuning improves translation quality even for zero-shot languages on average, which is impressive.
3. The paper further finds out that the improves are not defined by the language similarity, which leaves rooms for the future research on this direction.

**Reasons To Reject:**

1. The paper seems to be written in hurry. Table 2 (ref in Section 2) seems to be missing. Figure 5 is cropped.
2. The innovation of the paper seems to be limited. However, it is ok for an empirical paper.

---

> ### Author Rebuttal · Authors · 2024-05-30
>
> Thank you for your comments on the paper.
> We have fixed the formatting errors you mentioned.
> It would indeed be interesting to assess how trends hold with different LLMs.  We focused our work on the TOWER model family since at the time of writing it was one of the stronger contenders in comparison to the NLLB supervised multilingual MT system.

---

> > ### Comment · Reviewer_zRXH · 2024-06-05
> > **Acknowledge**
> >
> > Thanks for your response. I will keep my score.

---

### Official Review · Reviewer_rg12 · 2024-05-12

**Rating:** 7
**Confidence:** 4
**Ethics Flag:** 1

**Summary:**

The authors conduct a large scale empirical evaluation of the translation performance of the Tower language models, which are multilingual foundation models created from the LLama2 line of base models, and targeted at translation and related tasks such as quality estimation and automatic post editing. The evaluation measures translation performance in supervised and zero-shot settings, before and after fine-tuning. As a baseline and semi-oracle, the NLLB massively multilingual model is also evaluated in all settings.

**Reasons To Accept:**

- clearly written paper with very useful evaluation on partial- and fully-unsupervised MT tasks.
- in particular, the inclusion of best and worst performance in each setting will be useful for many researchers
- concrete takeaways, such as that pretraining on translation tasks benefits translation performance on unseen language pairs, but not for all language pairs uniformly/

**Reasons To Reject:**

- limited scientific novelty, in that the work is only a collection of evaluations

---

> ### Author Rebuttal · Authors · 2024-05-30
>
> Thank you for your comments on the paper.
> As you noted, this paper exclusively contributes a collection of evaluations to analyze model performance in multilingual settings beyond coarse averaging of metrics.  We believe that providing such a deep dive is critical to complement prior work which only reports average results and might thus obscure large gaps in performance between different language pairs.  We hope that this analysis can promote more comprehensive evaluations in future work, which can in turn guide the design of techniques that improve generalization across diverse language pairs.

---

### Decision · Program_Chairs · 2024-07-10

**Decision:**

Accept

**Comment:**

There appears to be a consensus among the reviewers. The paper presents a thorough evaluation of the TOWER family of language models, LLaMas specifically fine-tuned for MT. It draws several interesting and somewhat surprsing conclusions; notably, fine-tuning an LM for translation not only enhances performance in supervised language tasks but also improves results on languages / tasks not directly trained on. Additionally, there is considerable variability in performance across zero-shot languages. Although the reviewers acknowledge that the paper does not make a substantial technical contribution, they believe that the findings of this large scale study (132 translation directions!) are valuable to the community.